# Effect of 3-Dimensional Robotic Therapy Combined with Electromyography-Triggered Neuromuscular Electrical Stimulation on Upper Limb Function and Cerebral Cortex Activation in Stroke Patients: A Randomized Controlled Trial

**DOI:** 10.3390/bioengineering11010012

**Published:** 2023-12-22

**Authors:** Seo-Won Yang, Sung-Ryong Ma, Jong-Bae Choi

**Affiliations:** 1Department of Occupational Therapy, Sangji University, 83 Sangjidae-gil, Wonju-si 26339, Republic of Korea; solution1225@naver.com; 2Department of Occupational Therapy, Chosun University, 309 Pilmun-daero, Dong-gu, Gwangju 61452, Republic of Korea; masung79@chosun.ac.kr

**Keywords:** stroke, 3D-based robot therapy, electromyography-triggered neuromuscular electrical stimulation, cerebral cortex activation, upper-limb function

## Abstract

(1) Background: This study investigated the effect of 3-dimensional robotic therapy (RT) combined with electromyography-triggered neuromuscular electrical stimulation (RT–ENMES) on stroke patients’ upper-limb function and cerebral cortex activation. (2) Methods: Sixty-one stroke patients were assigned randomly to one of three groups. The stroke patients were in the subacute stage between 2 and 6 months after onset. The three groups received 20 min of RT and 20 min of electromyography-triggered neuromuscular electrical stimulation (ENMES) in the RT–ENMES group (n = 21), 40 min of RT in the RT group (n = 20), and 40 min of ENMES in the ENMES group (n = 20). The treatments were for 40 min, 5 days per week, and for 8 weeks. Upper-extremity function was evaluated using the Fugl–Meyer assessment for upper extremity (FMA-UE), Wolf motor function test, and action research arm test (ARAT); cerebral cortex activation and motor-evoked potential (MEP) amplitude were evaluated before and after the study. (3) Results: The analysis showed significant changes in all evaluation items for all three groups in the before-and-after comparisons. Significant changes were observed in the FMA-UE, ARAT, and MEP; in the posttest, the RT–ENMES group showed more significant changes in the FMA-UE, ARAT, and MEP than the other two groups. (4) Conclusions: The study analysis suggests that RT–ENMES effectively improves upper-limb function and cerebral cortex activation in patients with stroke.

## 1. Introduction

Patients with stroke generally show hemiplegia on the damaged hemisphere’s contralateral side and complex functional impairments, including spasticity, motor dysfunction, cognitive impairment, visual–perceptual impairment, and aphagia [1]. These disorders cause motor-control problems and are accompanied by upper-extremity muscle strength, stiffness, and sensory impairment [2]. More than 85% of patients with stroke experience hemiplegia, and >70% have upper-limb function impairment [3]. Among patients with damage to upper-extremity function, approximately 5% show normal recovery, and 20% recover some upper-extremity function. Functional recovery of the upper extremities becomes more difficult as patients with stroke enter the chronic stage; therefore, upper-extremity recovery is an important goal in treating patients with stroke [4].

Impaired upper-extremity function in patients with stroke limits the ability to use the arm or hold and manipulate objects, providing a significant barrier to the patient’s independent daily life and return to society, ultimately lowering their quality of life [5]. Therapeutic approaches to improve upper-extremity function in patients with stroke are being implemented in various ways; these interventions are based on neuroplasticity [6]. The recovery of upper-limb function in patients with stroke is closely related to intensive upper-limb practice with active neuromuscular activation through one’s own efforts [7]. Among the various treatment techniques used to restore upper-limb function in patients with stroke, electromyography-triggered neuromuscular electrical stimulation (ENMES) is the most common. ENMES stimulates muscles through an electric current, activating specific muscles to generate upper-limb movements, restore motor function, trigger sensory feedback to the brain during muscle contraction, and promote motor relearning. It also contributes to improved muscle strength [8,9]. ENMES may also limit the problem of “learned non-use” in which patients with stroke gradually become accustomed to managing daily activities without using specific muscles, considered an important barrier to maximizing motor-function recovery after stroke [10]. It was reported that a single ENMES treatment was effective in improving upper-limb function in patients with subacute stroke hemiplegia [11]. ENMES is an effective treatment that improves activities of daily living by improving the stretching and grasping functions of the paralyzed upper extremities. However, other studies have highlighted disadvantages of ENMES [12]. Difficulties may arise when NMES is used alone to activate multiple muscle groups for functional activity. NMES makes it difficult to control the contraction rate of individual muscles for upper-limb movements with the desired kinematic properties, including speed, trajectory, and movement smoothness, primarily because of muscle contractions evoked during electrical stimulation [13]. In addition, it may not be effective in patients who lack concentration and are interested in participating in the treatment. Recent research has addressed these shortcomings using 3-dimensional (3D)-based robotic therapy (RT) as a new treatment for patients with stroke. RT can improve concentration and motivation for treatment, and it is widely used in patients with limited upper-extremity movement [14]. RT can provide external auxiliary support for the upper extremities and help patients experience preprogrammed upper-extremity movements on the paretic side to improve the associated sensorimotor functions through repetitive practice [15]. RT is an innovative movement-based therapy that implements highly repetitive, intensive, adaptive, quantifiable, and task-specific arm training with feedback and motivation to enhance brain neuroplasticity [16,17,18]. Unlike humans, robotic devices programmed to perform in multiple functional modes ease the burden on rehabilitation providers and resource shortages without causing fatigue [19]. A 2018 Cochrane review found that electromechanical and robot-assisted arm training improved arm strength, arm function, and the performance of activities of daily living without increasing dropout rates or intervention-related adverse events compared with a variety of traditional treatment interventions [20].

Rehabilitation treatment using robots is an ideal tool for evaluating the movement patterns of each joint of the shoulder, elbow, wrist, and hand through dynamic measurements; it is controllable, repeatable, and quantifiable [21]. However, robotic systems use motors to provide external assistive torque to the limbs and do not have the same effect as ENMES, which generates movement by directly activating an individual’s specific muscles. In addition, activating specific muscle groups involved in the detailed joint movements of the upper extremities is limited. If the patient relies only on the robot’s movements, the individual may not make an effort to participate in the movements [22]. Currently, ENMES and RT are used separately in most rehabilitation treatments. Their combined effect on post-stroke paralyzed neuromuscular systems and rehabilitation has not been well evaluated. Treatment plans combining ENMES and RT must be justified to achieve optimized training effects because of each technique’s advantages and disadvantages [23]. This study aimed to quantify the complex effects of 3D-based upper-limb RT combined with ENMES on upper-limb function and cerebral cortex activation. In addition, we present evidence for a new treatment method for improving upper-extremity function in patients with hemiplegia after stroke.

## 2. Materials and Methods

### 2.1. Participants

The study participants were 69 subacute patients in the recovery stage within 6 months of stroke onset hospitalized at H Rehabilitation Hospital in Gyeonggi-do between January 2023 and June 2023. The subjects were patients diagnosed with stroke hemiplegia by a rehabilitation medicine doctor and were in the subacute phase 2 to 6 months after the onset of the disease. The evaluation and interviews in the process of selecting subjects to participate in the study were conducted by two occupational therapists with more than 10 years of experience. This study targeted patients who understood the purpose and content of this study and showed an active willingness to participate; informed consent was obtained from all patients. The sample size was set to 69 participants for the mean comparison (F-test) of the three groups using G-Power 3.1 with a significance level of 0.05, power of 0.9, and effect size of 0.25 [24]. To minimize selection bias, 23 people were randomly divided into three groups, the experimental group and control groups 1 and 2, using a computer random number table program. Figure 1 shows the Consolidated Standards of Reporting Trials (CONSORT) diagram for participant recruitment. This study was conducted according to the guidelines of the Declaration of Helsinki and approved by the Institutional Review Board of Chosun University (2-1041055-AB-N-01-2023-35).

The inclusion criteria were (1) adults >19 years of age, (2) patients with subacute hemiparesis <6 months after stroke onset, (3) patients capable of following instructions with a Mini-Mental State Test-Korea version score of ≥24, (4) patients with wrist extensor manual muscle test grade ≤3 (F), and (5) patients whose stiffness in the upper extremity on the affected side is grade ≤2 on the modified Ashworth scale. The exclusion criteria were (1) attachment of an artificial pacemaker, (2) patients with aphasia who have difficulty communicating, (3) patients with severe pain in the upper extremity on the paralyzed side (visual analog scale score of ≥5), (4) cases of peripheral nerve damage, skin lesions, or electrical hypersensitivity of the wrist extensor muscles on the affected side, and (5) because this study targeted patients with stroke, other vulnerable patients were excluded, including pregnant women and infants/children.

### 2.2. Study Procedure

This study was a single-blind, randomized, controlled trial using a three-group pretest–posttest design. All experiments and evaluations were conducted by two occupational therapists. The experiment for all three groups was conducted by an occupational therapist with >10 years of clinical experience. All evaluations were conducted by another occupational therapist with >10 years of clinical experience. This study divided 69 hospitalized patients randomly into three groups according to the order of visits using a computer-based random number table. The three groups received traditional rehabilitation treatment for 30 min a day, 5 times a week, and for 8 weeks. During the same period, the experimental group received ENMES and 3D-based upper RT for 20 min each (40 min total); control group 1 received 3D-based upper RT for 40 min, and control group 2 underwent an additional 40 min of ENMES treatment. The improvement of upper-extremity function was evaluated using the Fugl–Meyer assessment for upper extremity (FMA-UE), Wolf motor function test (WMFT), and action research arm test (ARAT). Cerebral cortex activation was evaluated by using the motor-evoked potential (MEP) amplitude, measured using transcranial magnetic stimulation.

### 2.3. Intervention

#### 2.3.1. Electromyography-Triggered Neuromuscular Electrical Stimulation (ENMES)

This study used an EMG FES 2000 (Walking Man II, Iksan, Republic of Korea) as the ENMES.

Three surface electrodes were placed on the wrist extensor muscles, extensor pollicis brevis, and extensor pollicis longus (Figure 2 and Figure 3). First, voluntary wrist extension was induced, and a reference threshold was set according to the level of action potential due to muscle contraction. When the action potential reached the reference threshold and electrical stimulation was induced, 0.1 s rise-phase, 5 s contraction-phase, and 2 s fall-phase processes were applied using 35 Hz, a pulse width of 200 µs, and a symmetric rectangular biphasic signal. The stimulation intensity was set between 15 and 30 mA. If the action potential generated through muscle contraction did not reach the reference threshold, electrical stimulation was set to appear automatically after 20 s; the reference threshold setting was reset for each treatment session [25].

#### 2.3.2. 3D-Based Robotic Therapy (RT)

The 3D-based upper-limb RT used in this study was the ReoGo-J (ReoGoTM; Motorika Medical Ltd., Caesaria, Israel). This end-effector robotic system activates moments in the paralyzed shoulder, elbow, and forearm. During robotic training, the patients performed several tasks at an assistance level appropriate for their functional level, including forward reaching, abduction, and external rotation. Using a secondary controller, such as an active secondary controller, ReoGo-J allows for patients with stroke to move their damaged upper extremities independently [26]. The accuracy of the performance was aided by visual feedback from the ReoGo-J to the patient through a front-facing monitor. The mobility of the shoulder, elbow joints, and forearm allows for specific treatment of the upper limbs. Robots enable the execution of movements in 3D and spatial planes. Exercises can be performed variously using the forearm, wrist, or handgrips. Thus, the system allows for users to perform different exercises to reach their goals through visual and auditory feedback on a connected computer screen [27]. Movement modes can vary from passive to active with different levels of intervention that the patient exerts on the robotic arm. The movement’s range of motion can be adjusted according to the unique characteristics of each participant. The range of motion was measured and set according to the patient’s personal upper-extremity function level; training was then conducted to improve movement through assistance in areas outside the range. Of 71 performed tasks, 10 were selected and applied in this study. The tasks involved forward reaching (2D) and forward reaching (3D). Abduction reaching, radial reaching (2D), radial reaching (3D), reaching in eight directions, reaching for the mouth, reaching for the head, and game mode (puzzle, kitchen) were selected and performed according to the patient’s functional level [28]. The experimental group applied it for 20 min a day; control group 1 applied it for 40 min, 5 days a week, and for 8 weeks (Figure 4).

### 2.4. Outcome Measures

#### 2.4.1. Fugl–Meyer Assessment for Upper Extremity (FMA-UE)

The Fugl–Meyer assessment (FMA) evaluates motor function on the paralyzed side of patients with stroke based on Brunnstrom’s six-step recovery process. This study evaluated only the upper-extremity items of the FMA (FMA-UE) comprising 33 items, including 18 items from the shoulder, elbow, and forearm; 5 items from the wrist; 7 items from the hand and fingers; and 3 items measuring coordination. The score is on a 3-point scale from 0 to 2; points are awarded depending on whether the performance is completed. A score of 0 indicates impossible to perform, 1 indicates partial performance, and 2 indicates complete performance. The mean total score for upper-extremity function is 66 points. The inter- and intra-rater reliability of the FMA upper-extremity test was very high (0.97) [29].

#### 2.4.2. Wolf Motor Function Test (WMFT)

The Wolf motor function test (WMFT) was developed in 1989 to evaluate upper-extremity motor function in patients with hemiplegia. The test measures each activity’s exercise performance and performance time and consists of 17 movement tasks that range from simple to complex. The score is on a 6-point scale ranging from 0 to 5 with 0 indicating no performance and 5 indicating normal movement; lower scores indicate worse motor performance. The inter-rater reliability of this tool’s function score was 0.88; the performance time was 0.97 [30].

#### 2.4.3. Action Research Arm Test (ARAT)

The action research arm test (ARAT) assesses the ability to perform gross movements of the upper extremities and grasp, move, and release objects of various sizes, weights, and shapes. ARAT is an evaluation tool that evaluates upper-extremity function and release ability, and its development is based on Carroll’s upper-extremity function test [31,32]. It consists of four sub-items with a total of 19 items, including grasp (6 items), grip (4 items), pinch (6 items), and gross movements (3 items). On a 4-point scale (0–3), impossible to perform is 0 points, partial performance is 1 point, performing the test fully but taking a long time or showing difficulties is 2 points, and performing the test normally and completely is 3 points. The total score is 57 points: 0 points for no movement and 57 points for performing all movements without difficulty. The intra-tester reliability of the ARAT was 0.99; the test–retest reliability was 0.98 [32].

#### 2.4.4. Motor-Evoked Potential (MEP) Amplitude

The motor-evoked potential (MEP) amplitude was measured using the Nicolet Viasys Viking Select EMG EP system (San Diego, CA, USA). The MEP is an objective electrodiagnostic evaluation tool that induces specific peripheral muscle responses through transcranial magnetic stimulation of the cerebral motor cortex. For magnetic stimulation, the International Electroencephalograph 10–20 recording method was applied; the central part of the coil stimulator was placed at the Cz position. The subjects were placed in the supine position in an isolated space with the center of the coil contacting the cerebral hemisphere on the unaffected side. The MEP evaluation was conducted by a rehabilitation medicine doctor to ensure safety, and the subject’s vital signs were monitored during the evaluation. The first dorsal interosseous (FDI) muscle was located in the motor cortex at a 45° angle from the centerline and was moved gradually to determine the point of maximum response. The maximum magnetic field strength was 2.0 Tesla; the stimulation time was 0.1 ms [33]. The stimulation intensity was increased gradually from 80% to 100%, and the stimulation was performed multiple times. EMG values were measured by attaching a silver–silver chloride electrode to the FDI muscle on the affected side using the belly-tendon method and a ground electrode to the arm [34]. The resting motor threshold was defined as the minimum stimulation intensity at which MEPs > 50 μV were recorded at least 5 times during 10 stimulations. The MEP amplitude was determined by measuring the amplitude 12 times after 120% stimulation [35]. The peak-to-peak amplitudes of the evoked MEPs from the contralateral target muscles were obtained. The inter-stimulus interval in our study was approximately 5 s to minimize carry-over effects of the previous stimuli. EMG values were recorded using the mobile Viking Select software 19.1; signals were amplified at 100 ms/div and filtered from 2 Hz to 10 KHz.

### 2.5. Statistical Analysis

The data collected in this study were statistically analyzed using SPSS (version 22.0; SPSS Inc., Chicago, IL, USA). Baseline variables were compared between groups using one-way analysis of variance (ANOVA) and the Kruskal–Wallis or Fisher’s exact tests, depending on the characteristics of the variables. A paired *t*-test was used to compare the average changes in upper-extremity function and cerebral cortex activation before and after the intervention in the three groups. One-way ANOVA was used to compare the average changes in upper-limb function and cerebral cortex activation before and after the experiment and the amount of change among the three groups. A post hoc test was performed (assuming equal variance) using the Scheffe method; if an equal variance was not assumed, Dunnett’s T3 method was used. All statistical significance levels were set at α = 0.05.

## 3. Results

### 3.1. Participant Characteristics

The general characteristics of the participants are presented in Table 1. A homogeneity test was conducted on all items among the three groups; no significant differences were observed (Table 1).

### 3.2. Comparison between the Experimental and Control Groups

In the before-and-after comparison of the three groups, all groups showed significant changes in the FMA-UE, WMFT, and ARAT, which are evaluations of upper-extremity function, and MEP, which is an evaluation of cerebral cortical activation. The pre- and post-comparison between the three groups showed significant changes in the FMA-UE, ARAT, and MEP, and the post hoc test of the three evaluation items showed significant results in the comparisons of the RT–ENMES and RT groups and the RT–ENMES and ENMES groups (Table 2).

### 3.3. Changes in the Groups before and after Intervention

The comparison between the three groups showed significant changes in the FMA-UE, ARAT, and MEP and the post-hoc test of the three evaluation items. Significant results were also found in the comparison of the RT–ENMES and RT groups and the RT–ENMES and ENMES groups (Table 3, Figure 5).

## 4. Discussion

Recovery of motor function after stroke is slower in the upper extremities than in the lower extremities; hand function recovery is among the slowest [36]. Therefore, the recovery of upper-extremity function is an important goal in rehabilitation treatment, and many therapeutic methods and approaches are being attempted in clinical practice for this purpose. This study combined ENMES and 3D-based upper-limb RT to investigate the effect on the recovery of upper-limb function and cerebral cortex activation in patients with stroke. A before-and-after comparison of the three groups in this study showed significant changes in upper-extremity function and cerebral cortex activation. This finding is consistent with many studies that showed positive effects from RT and ENMES interventions applied singly or combined on upper-limb function and brain activation in patients with stroke [10,11,12,14,15,16,17,18]. However, a comparison of the three groups revealed differences. A significant change was found in the pre- and post-average comparisons among the three groups in the FMA-UE and ARAT, which are upper-extremity function evaluations, but no significant change in the WMFT. In the post hoc tests of the FMA-UE and ARAT evaluations, the RT–ENMES group showed significant changes compared to the RT and ENMES groups. In addition, when comparing the change in the upper-extremity function evaluation between the three groups, the same significant change was shown in the FMA-UE and ARAT evaluations. In the post hoc test, the RT–ENMES group showed a significant change compared to the RT and ENMES groups.

RT and ENMES are more effective in improving upper-extremity function in patients with stroke when combined as a single intervention than when administered alone. Combining the two interventions improved upper-extremity function effectively, meaning that patients could make precise movements by controlling the specific muscle groups necessary for functional use. This factor appears important for ensuring that patients have a kinesthetic experience with the movements to be learned [37]. RT repeatedly assisted upper-limb movements through external power; ENMES improved the kinesthetic experience of stimulated wrist extensor muscles. Therefore, the combined approach of RT and ENMES may bring additional benefits to upper-limb recovery [22]. Important factors in improving upper-extremity function in patients with stroke include the willingness to participate in treatment, motivation, and interest. Parallel RT and ENMES interventions were related directly to these factors. ENMES is more effective than general NMES as an active treatment in which the patient participates through voluntary effort and motivation [38,39]. Repetitive activities for voluntary motivation and afferent stimulation are effective for the neurological recovery of the paralyzed upper extremities [40]. The 3D-based RT provides real-time feedback on upper-limb movements through a 3D computer screen, effectively improving concentration and movement coordination. The performance tasks were also continuous goal-oriented tasks; the participants could voluntarily participate in the intervention in a more interesting way because they chose the tasks they wanted and performed them in game mode among various tasks [41]. The advantages of these two interventions are believed to combine.

Three assessment tools were used to evaluate the changes in upper-limb function. However, no significant difference was found between the groups on the WMFT; the results appeared to differ depending on the difficulty of performing the evaluation tool. Compared with the FMA-UE, the WMFT places more weight on items evaluating detailed hand movements and manipulative abilities. Because the study participants comprised patients with moderate impairment, the WMFT evaluation partially confirmed differences in upper-extremity function [42]. In addition, the FMA-UE and ARAT correlate highly in the evaluation of upper-extremity function in patients with moderately impaired stroke [43].

MEP, which evaluates brain activation, changed significantly among the three groups; a significant change was confirmed in the post hoc test when the RT–ENMES group was compared with the RT and ENMES groups. This change is believed to have affected brain neuroplasticity and reorganization of the areas related to upper-limb function in the RT–ENMES group and may have contributed to the positive response to upper-limb functional use. Both RT and ENMES affect brain neuroplasticity. ENMES activates the motor nerve pathway from the peripheral nervous system to the central nervous system through muscle contraction on the paretic side. Fujiwara et al. showed reciprocal inhibitory modulation of short intracortical inhibition and finger extensor muscles resulting from a single NMES intervention, supporting the results of the present study [44]. RT is thought to enhance motor-nerve activation by providing additional systematic and repetitive movements [45]. Therefore, the parallel intervention of the two treatments was effective in recovering upper-limb movement through a positive synergistic effect on brain neuroplasticity in patients with stroke. The 3D-based RT provides visual feedback and immersion, allowing for patients to participate more effectively in rehabilitation. ENMES provides direct motor feedback by inducing muscle contraction. A recent NMES study showed that NMES training targeting upper-extremity function in chronic stroke patients induced modulation of somatosensory-evoked potentials accompanying sensory recovery [46]. Combining these two feedbacks results in the interaction of the sensory-motor system, leading to an overall improvement in the MEP [20]. One limitation of this study is that it targeted patients in the subacute stage of stroke; therefore, a natural recovery effect is expected. The treatment effect may vary depending on factors that include the severity of the stroke, age, sex, side of the injury, and disease-onset period; therefore, additional research is necessary. Although improvements in cerebral cortex activation and upper-extremity function have been reported, actual changes in daily living activities have not been evaluated. The research period was short at eight weeks; since no lasting effects were confirmed after eight weeks of research, this short period should be considered in future research.

## 5. Conclusions

This study showed that the combined intervention of ENMES and 3D-based upper-limb RT effectively improved upper-limb function and cerebral cortex activation in patients with stroke. This study provides a scientific basis for proposing a new concurrent intervention method to improve upper-limb function in patients with stroke.

## Figures and Tables

**Figure 1 bioengineering-11-00012-f001:**
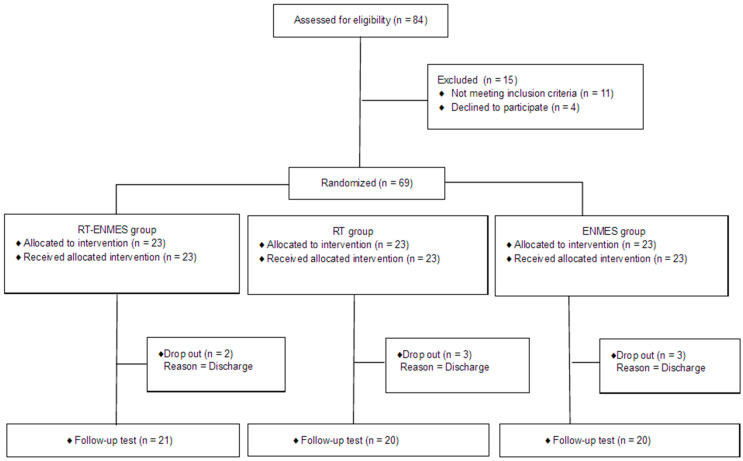
CONSORT diagram of participant recruitment.

**Figure 2 bioengineering-11-00012-f002:**
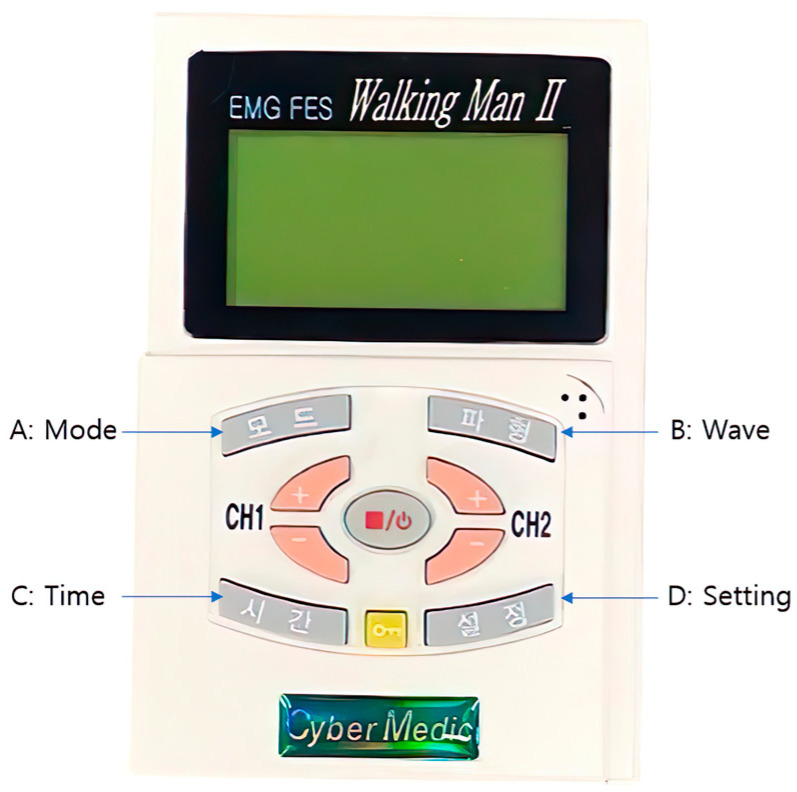
EMG FES 2000 (Walking man Ⅱ, Iksan, Republic of Korea).

**Figure 3 bioengineering-11-00012-f003:**
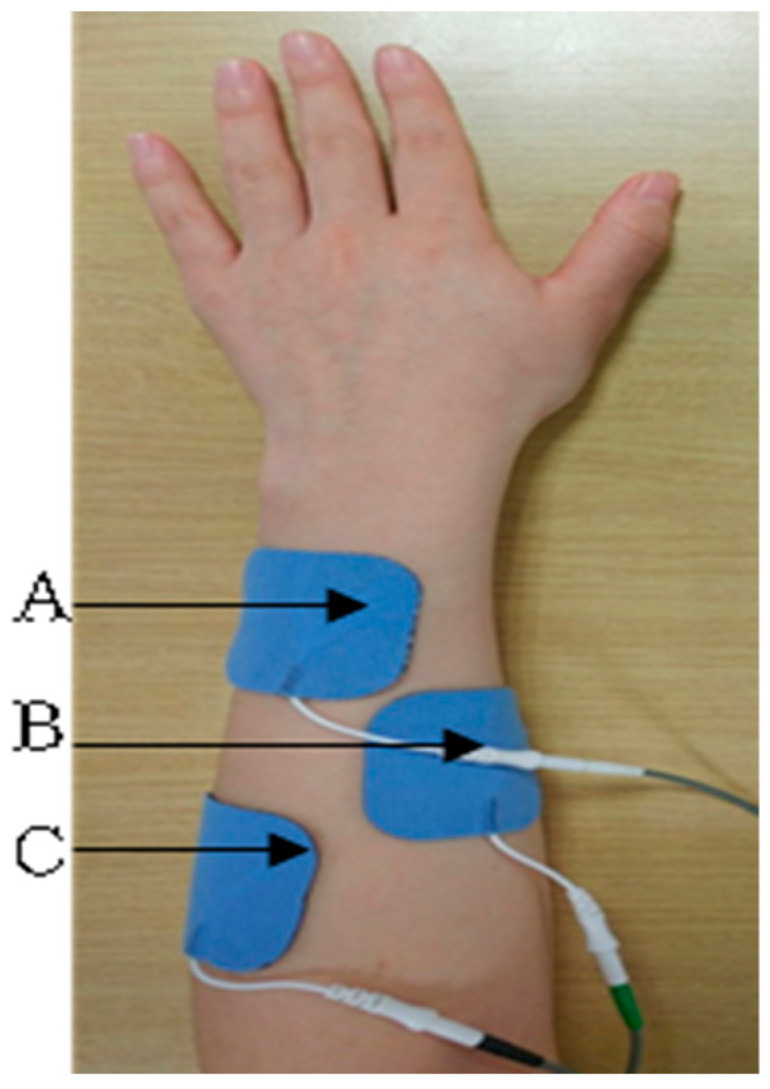
The attached surface electrodes of ENMES. A and C: active electrode and the reference electrode at the origin and insertion sites of the extensor pollicis brevis and extensor pollicis longus. B: EMG electrode.

**Figure 4 bioengineering-11-00012-f004:**
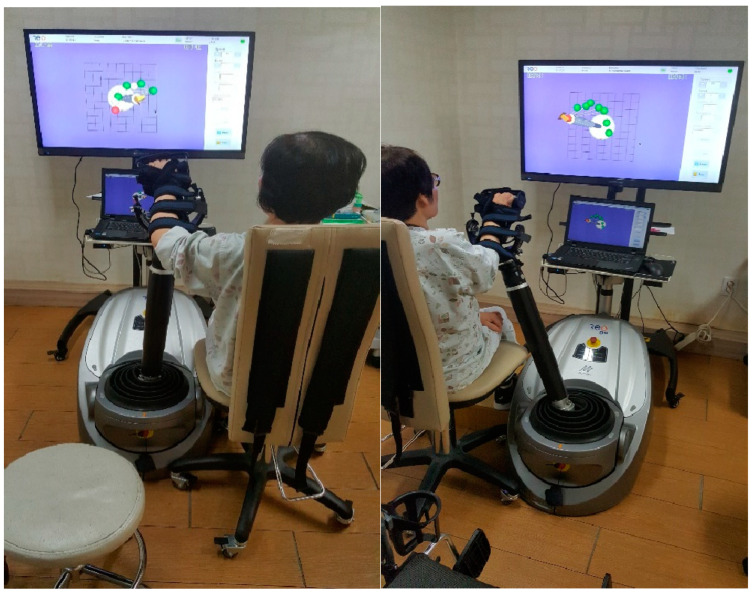
The 3-dimensional-based robotic therapy.

**Figure 5 bioengineering-11-00012-f005:**
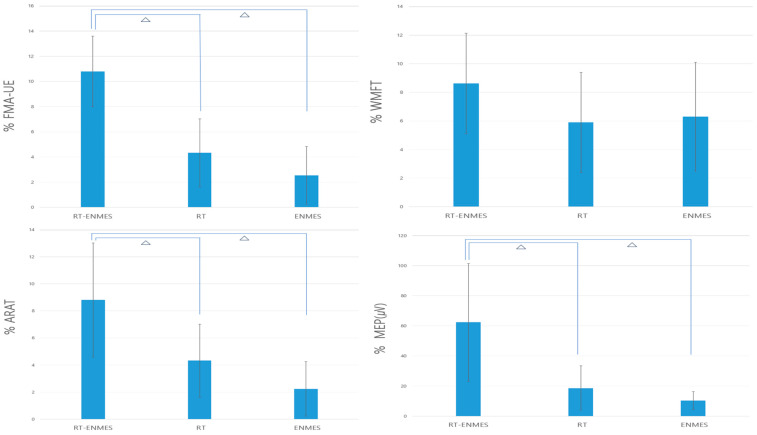
Changes in both groups before and after the intervention (^ᅀ^ *p* < 0.05 with post-hoc test (P1 = RT–ENMES − RT; P2 = RT–ENMES − ENMES; P3 = RT−ENMES).

**Table 1 bioengineering-11-00012-t001:** Characteristics of participants.

Characteristics	RT–ENMES Group (n = 21)	RT Group (n = 20)	ENMES Group (n = 20)	F	*p*
Age (year), mean ± SD	61.10 ± 7.66	63.38 ± 8.34	63.10 ± 8.57	0.483	0.619
Gender (male/female)	10/11	12/9	10/10	0.197	0.821
Type of stroke (Hemorrhage/Infarction)	7/14	9/12	9/11	0.964	0.387
Side of stroke (Right/Left)	11/10	8/13	11/9	0.669	0.516
Time since onset of stroke months, mean ± SD	3.48 ± 1.12	3.67 ± 1.23	3.60 ± 1.31	0.130	0.878

RT: robotic therapy; ENMES: electromyography-triggered neuromuscular electrical stimulation; SD: standard deviation.

**Table 2 bioengineering-11-00012-t002:** Comparison between the experimental and control groups.

	RT–ENMES Group		RT Group		ENMES Group		*p*	Post hoc Test
	Before Treatment	After Treatment	Before Treatment	After Treatment	Before Treatment	After Treatment		
FMA UE	16.76 (3.80)	27.57 (4.99) **	15.33 (4.85)	19.67 (4.83) **	15.95 (3.94)	18.50 (4.60) **	0.000 ^†^	P1 = 0.000 ^ᅀ^ P2 = 0.000 ^ᅀ^ P3 = 0.710
WMFT	15.14 (5.24)	23.76 (4.63) **	16.05 (6.62)	22.10 (6.92) **	16.60 (7.49)	22.90 (6.04) **	0.293	
ARAT	10.43 (3.37)	19.24 (3.16) **	10.90 (2.99)	15.24 (3.92) **	10.74 (3.33)	13.15 (4.34) **	0.000 ^†^	P1 = 0.004 ^ᅀ^ P2 = 0.000 ^ᅀ^ P3 = 0.259
MEP (μV)	132.10 (53.02)	194.47 (60.11) **	138.28 (40.57)	157.31(49.14) **	126.92 (49.62)	137.43(53.28) **	0.002 ^†^	P1 = 0.011 ^ᅀ^ P2 = 0.003 ^ᅀ^ P3 = 0.847

The values are mean ± standard deviation. RT: robotic therapy; ENMES: electromyography-triggered neuromuscular electrical stimulation; FMA UE: Fugl–Meyer assessment for upper extremity; WMFT: Wolf motor function test; ARAT: action research arm test; MEP: motor-evoked potential, ** *p* < 0.01 with paired *t*-test, ^†^ *p* < 0.05 with one-way ANOVA, ^ᅀ^ *p* < 0.05 with post hoc test (P1 = RT–ENMES − RT; P2 = RT–ENMES − ENMES; P3 = RT−ENMES).

**Table 3 bioengineering-11-00012-t003:** Changes in the groups before and after intervention.

	RT–ENMES Group	RT Group	ENMES Group	*p*	Post hoc Test
FMA UE	10.81 (2.80)	4.33 (2.70)	2.55 (2.44)	0.000 ^†^	P1 = 0.000 ^ᅀ^ P2 = 0.000 ^ᅀ^ P3 = 0.257
WMFT	8.62 (3.52)	5.90 (3.74)	6.30 (3.88)	0.056	
ARAT	8.81 (4.22)	4.33 (2.70)	2.25 (2.07)	0.000 ^†^	P1 = 0.002 ^ᅀ^ P2 = 0.000 ^ᅀ^ P3 = 0.081
MEP (μV)	62.36 (39.18)	18.80 (14.64)	10.50 (6.09)	0.000 ^†^	P1 = 0.000 ^ᅀ^ P2 = 0.000 ^ᅀ^ P3 = 0.798

The values are mean ± standard deviation. RT: robotic therapy; ENMES: electromyography-triggered neuromuscular electrical stimulation; FMA UE: Fugl–Meyer assessment for upper extremity; WMFT: Wolf motor function test; ARAT: action research arm test; MEP: motor-evoked potential, ^†^ *p* < 0.05 with one-way ANOVA, ^ᅀ^ *p* < 0.05 with post-hoc test (P1 = RT–ENMES − RT; P2 = RT–ENMES − ENMES; P3 = RT−ENMES).

## Data Availability

The datasets generated during the current study are available from the corresponding author upon reasonable request.

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
