# Peer review of "Effect of 3-Dimensional Robotic Therapy Combined with Electromyography-Triggered Neuromuscular Electrical Stimulation on Upper Limb Function and Cerebral Cortex Activation in Stroke Patients: A Randomized Controlled Trial"

_bioengineering, 2023, doi:10.3390/bioengineering11010012_

Round 1
Reviewer 1 Report
Comments and Suggestions for Authors
Upper extremity paresis constitutes a consequential impairment observed in stroke patients, impeding their autonomy in activities of daily living (ADL), reintegration into the workforce, and societal participation, thereby constricting their overall quality of life (QOL). In this randomized controlled trial (RCT), the investigators scrutinized the impact of a combinatory therapeutic approach involving 3-Dimensional robotic therapy and electromyography-triggered neuromuscular electrical stimulation (ENMES), particularly in terms of motor functional recovery, in comparison to the effects of each singular treatment modality. The authors demonstrated that the combinatory treatment engendered the most substantial improvement in upper extremity function, as evaluated through assessments such as the Fugl-Meyer upper extremity function, Action Research Arm test, Wolf Motor Function Test, and Motor Evoked Potentials. Although the present approach will have a potential to promote the feature of stroke rehabilitation, an examination of the data interpretation and discussion may not be sufficiently done. The study methodology, in particular, raises noteworthy concerns, as articulated by the reviewer.
Major concern
No control group is included.
Although the authors describe the rehabilitation system “ReoGo-J”, the hand module of this device is attached to a pole. Thus, the movability is basically on the spherical surface, which is rather 2D than 3D. When the length of the pole can be changed as smoothly as in front-back and right-left directions, it can be 3D system. Was it possible? Otherwise, the description “3D” will be an overstatement and must be amended.
Abstract: The phase of stroke influences the effect of treatment to a large extent. Please clearly explain the phase of stroke treated in this study.
Introduction:
As far as the reviewer knows, some studies showed the treatment effect of single NMES treatment targeting upper extremity paresis in subacute stroke patients as in the authors setting; the following is the representative: Shindo K et al., Effectiveness of Hybrid Assistive Neuromuscular Dynamic Stimulation Therapy in Patients With Subacute Stroke: A Randomized Controlled Pilot Trial. Neurorehabilitation and Neural Repair 25(9) 830–837. Introduction should be included regarding such studies.
Material and methods
Page 3 lines 105-
It is unclear when the patients are eligible to participate in the experiment. It is also unclear how frequently and by whom the eligibility was assessed for the subacute stroke inpatients.
Page 3 lines 122-
How did the authors treat participants with severe aphasia or cognitive impairment? Are the tasks easy enough for those patients to implement correctly?
P7 lines 247-259
Regarding the MEP assessment, please clarify how the authors ensure the safety of applying transcranial magnetic stimulation (TMS) on the impaired hemisphere.
Discussion
Page 12, lines 385-386
Though the authors mention that the MEP shows neuroplasticity and reorganization, it is highly variable. Only 12 stimuli could not represent neuroplasticity and reorganization sufficiently. In addition, other measurements using TMS, such as short- and long-intercortical inhibition and intercortical facilitation, must be needed.
It is unclear which ENMES or robotic training-induced neuroplasticity and cortical reorganization is not discussed in detail. Particularly the discussion “lines 387-389: Both RT and ENMES affect brain neuroplasticity. ENMES activates the motor nerve pathway from the peripheral nervous system to the central nervous system through muscle contraction on the paretic side.” is not sufficient. For example, Fujiwara showed a single NMES intervention-induced modulation of SICI and reciprocal inhibition of finger extensor muscles. (Fujiwara T et al., Motor improvement and corticospinal modulation induced by hybrid assistive neuromuscular dynamic stimulation (HANDS) therapy in patients with chronic stroke. Neurorehabil Neural Repair 2009 23: 125)
(lines 395-396) Though the authors indicate the importance of sensorimotor functioning, the cited article [44] focuses on almost only the motor system. A recent NMES study showed that NMES training targeting upper extremity function in chronic stroke patients induced modulation of somatosensory-evoked potentials accompanying sensory recovery. (Tashiro S et al., Neuromuscular electrical stimulation-enhanced rehabilitation is associated with not only motor but also somatosensory cortical plasticity in chronic stroke patients: an interventional study. Ther Adv Chronic Dis. 2019 Nov 20;10:2040622319889259.)
Discussion integrating those progress in NMES studies will contribute to improving the discussion of the present manuscript.
Minor concerns:
The topic of paragraphs 2-4 (Page 2, line 45- Page 3, line 99) seems not well integrated. The reviewer thinks that each introduction for ENMES and RT should be summarized separately, and the last paragraph should summarize combination matter.
Page 3 line 105 and Page 4 line 139
Though the authors describe the number of participants as 61, it must be 69, including the cases that dropped out of the study.
Page 5 line 153
Please check the indent of the line.
P7 lines 247-259
Regarding MEP assessment, the information on inter-stimulus interval seems missing.
Reviewer 2 Report
Comments and Suggestions for Authors
The conclusion and methodology does not support evidence of specific areas of cortical activation
suggest focusing the discussion on potential activation
Did you have cortical electrodes to measure brain function or functional imaging to show activation amd did you account for the robotic passive movement being a passive source of cortical activity
agree that therapy is important to recovery
include limitations
a good effort
Round 2
Reviewer 1 Report
Comments and Suggestions for Authors
The reviewer considers that the manuscript is acceptable in the present form.
Reviewer 2 Report
Comments and Suggestions for Authors
Authors addressed areas sufficiently